# Computational Tool to Design Small Synthetic Inhibitors Selective for XIAP-BIR3 Domain

**DOI:** 10.3390/molecules28135155

**Published:** 2023-06-30

**Authors:** Marc Farag, Charline Kieffer, Nicolas Guedeney, Anne Sophie Voisin-Chiret, Jana Sopkova-de Oliveira Santos

**Affiliations:** Normandie Univ., UNICAEN, CERMN, 14000 Caen, France; marc.ragui@unicaen.fr (M.F.); charline.kieffer@unicaen.fr (C.K.); nicolas.guedeney@univ-rennes.fr (N.G.); anne-sophie.voisin@unicaen.fr (A.S.V.-C.)

**Keywords:** XIAP-BIR3, synthetic inhibitor, molecular dynamics, MM-PBSA free energy prediction, 3D Pharmacophore

## Abstract

X-linked inhibitor of apoptosis protein (XIAP) exercises its biological function by locking up and inhibiting essential caspase-3, -7 and -9 toward apoptosis execution. It is overexpressed in multiple human cancers, and it plays an important role in cancer cells’ death skipping. Inhibition of XIAP-BIR3 domain and caspase-9 interaction was raised as a promising strategy to restore apoptosis in malignancy treatment. However, XIAP-BIR3 antagonists also inhibit cIAP1-2 BIR3 domains, leading to serious side effects. In this study, we worked on a theoretical model that allowed us to design and optimize selective synthetic XIAP-BIR3 antagonists. Firstly, we assessed various MM-PBSA strategies to predict the XIAP-BIR3 binding affinities of synthetic ligands. Molecular dynamics simulations using hydrogen mass repartition as an additional parametrization with and without entropic term computed by the interaction entropy approach produced the best correlations. These simulations were then exploited to generate 3D pharmacophores. Following an optimization with a training dataset, five features were enough to model XIAP-BIR3 synthetic ligands binding to two hydrogen bond donors, one hydrogen bond acceptor and two hydrophobic groups. The correlation between pharmacophoric features and computed MM-PBSA free energy revealed nine residues as crucial for synthetic ligand binding: Thr308, Glu314, Trp323, Leu307, Asp309, Trp310, Gly306, Gln319 and Lys297. Ultimately, and three of them seemed interesting to use to improve XIAP-BR3 versus cIAP-BIR3 selectivity: Lys297, Thr308 and Asp309.

## 1. Introduction

Apoptosis escape is one of the major causes of cancer development and progression. It also contributes to chemoresistance [1]. Restoring apoptosis in cancer cells is therefore a promising strategy for the new anticancer therapy development. X-linked inhibitor of apoptosis protein (XIAP), also known as “inhibitor of apoptosis protein 3” (IAP3) or “baculoviral IAP repeat-containing protein 4” (BIRC4), is a protein inhibiting cell apoptosis. To stop apoptotic cell death, XIAP binds to caspase-9, caspase-3 and caspase-7, three enzymes that are essential for apoptosis initiation and execution [2,3]. Binding to caspases allows XIAP to inhibit their activation to block intrinsic, as well as extrinsic, signaling apoptosis pathways. XIAP is overexpressed in many tumors [4], and its anti-apoptotic effect contributes to the cancer cell’s escape from apoptosis. Moreover, it plays an important role in chemoresistance and has therefore become an important target for the malignancy treatment [4].

XIAP belongs to the inhibitor of apoptosis proteins (IAPs) family, which consists of eight different proteins. They all share the zinc-binding baculovirus-IAP-repeat (BIR) domain comprising around 70 amino acids. Among IAP family members, XIAP is the only one that inhibits caspases by direct physical interactions. Two other members of this family, c-IAP1 and c-IAP2, are also able to bind caspase-3 and -7 but do not inhibit them through physical interaction. 

XIAP contains three BIR domains in the *N*-terminal region (BIR1, BIR2 and BIR3), an ubiquitin-associated (UBA) domain and a *C*-terminal region called zinc-binding Really Interesting New Gene (RING) domain [5]. To inhibit the caspase catalytic activity, XIAP uses different XIAP domains and different mechanisms. 

Caspase-9 inhibition by XIAP occurs through its BIR3 domain. Through its surface groove in Zn chelator proximity, XIAP-BIR3 domain traps caspase-9 in a monomeric inactive state. This way, it deprives caspace-9 of any catalytic activity. Four amino acids in caspase-9 *N*-terminal region small subunit (A_316_-T_317_-P_318_-F_319_) bind to the XIAP-BIR3 surface groove. In the case of caspase-3/-7 effector, BIR1-BIR2 linker peptide preceding XIAP-BIR2 interacts with caspase-3/-7 [6,7,8]. This XIAP BIR1-BIR2 linker segment directly occupies the caspase active site, resulting in a substrate entry blockage. Interestingly, although the XIAP BIR1-BIR2 linker sequence plays a dominant role in inhibiting caspase-3 and -7, this fragment in isolation is insufficient to provide an inhibitory effect [9,10]. Indeed, caspase-3 in a crystal structure also binds to XIAP-BIR2 domain surface groove in Zn chelator proximity [8].

In normal cells, second mitochondrial caspase activator (Smac) protein is a natural XIAP inhibitor. In response to an apoptotic stimulus, Smac protein is released by mitochondria; binds to XIAP-BIR3 and XIAP-BIR2 binding grooves; and removes IAP-mediated inhibition of caspase-3, -7 and -9. The released caspases can then exert their pro-apoptotic activity.

The XIAP-BIR3 domain solved structure complexed with a functionally active 9-residue peptide derived from the Smac *N*-terminus showed that interaction occurs principally through four residues located on Smac *N*-terminal region (A_1_-V_2_-P_3_-I_4_, called AVPI). AVPI binds to a surface groove on XIAP-BIR3, with the protonated *N*-terminal Ala1 making several hydrogen bonds with neighboring residues on XIAP-BIR3 [11,12]. Smac *N*-terminal binding groove in XIAP-BIR3 is the same as the one binding caspase-9, and Smac tetrapeptide has a sequence similarity with the caspase-9 binding *N*-terminus. Smac *N*-terminal peptide is structured in a beta strand [12,13]. Three intramolecular hydrogen bonds between Val2/Ile4 backbone in Smac and Gly306/Thr308 backbone in XIAP-BIR3 allow the formation of a four-stranded antiparallel β-sheet. Thus, the antiparallel β-sheet of three strands from the XIAP-BIR3 domain is completed during the complex formation by a fourth strand, Smac *N*-terminal AVPI peptide.

The XIAP-BIR3 domain was initially exploited as a potential target for anticancer drugs in overexpressing IAP tumors. Smac AVPI sequence was the basis of the initial work in XIAP antagonist field. These antagonists, called “Smac-mimetics”, have a high structural similarity with the endogenous AVPI peptide from Smac. Four main generations of small molecules have emerged, including peptides and peptidomimetics, conformational-constrained monovalent or bivalent antagonists. More recently, non-alanine-based antagonists have risen. Nevertheless, “Smac-mimetics” presented drawbacks, such as structural fragility. Thus, an effort was made to develop synthetic non-peptide antagonists. The XIAP-BIR3 antagonist development resulted in a few small molecule inhibitors that reached clinical evaluation [14,15]. Nevertheless, it was observed that XIAP-BIR3 antagonists lack selectivity for XIAP. Indeed, XIAP inhibitors interact with both cIAP-1 and cIAP-2 through their BIR3 domains, leading to a cIAP1-2 inhibition [16]. As a consequence, there is an increase of TNFα releasing, resulting in a serious side effect called cytokine release syndrome [17,18]. Cytokine release syndrome has caused the early termination of several clinical trials. Consequently, the development of optimized and selective antagonists for XIAP remains a challenge to overcome.

In this study, we focused on the setting up of an in silico tool that allowed us to design and/or optimize non-peptidic and non-peptidomimetic XIAP-BIR3 selective antagonists. To this end, we implemented and evaluated the efficiency of different molecular dynamic simulation strategies. Our strategies included a ranking of relative XIAP-BI3 antagonist binding affinities, using the Molecular Mechanics Poisson–Boltzmann Surface Area (MM-PBSA) approach. MM-PBSA is widely employed to predict ligand–receptor free energy thanks to its efficiency and reasonable accuracy. Then, we exploited the molecular dynamics simulations that were producing the best agreement with experimental binding data in order to build a 3D pharmacophore for a synthetic XIAP-BIR3 antagonist. The pharmacophore performance was assessed and optimized with a testing chemolibrary. The MM-PBSA data analysis correlated with the held features of our final optimized pharmacophore. This also allowed us to highlight the residues to target and increase XIAP-BIR3 vs. cIAP1/2-BIR3 selectivity.

## 2. Results and Discussion

### 2.1. MM-PBSA Binding Free-Energy Prediction

To reach our goal, we started by setting up a method for rapid and reliable predictions of synthetic ligand-binding affinities to the XIAP-BIR3 domain. To assess the predictive performance, we selected four XIAP-BIR3 synthetic ligands for which the X-ray structures of their complexes with XIAP-BIR3 (residues 248–352) were known, and the binding constants were measured experimentally. At the time of our work, 36 3D structures with XIAP-BIR3 were available in the PDB database [19]; 4 NMR solution structures and 32 structures were solved using X-ray diffraction. Among them, only 16 were co-crystallized with a non-peptide or non-peptidomimetic ligand. For our study, we selected four complexes from sixteen available ones with the following PDB IDs: 5C7C [20], 5M6M [21], 5OQW [22] and 5M6L [21]. The choice of these complexes was based on the ligand structural diversity/similarity, as well as on the difference/similarity in their binding activities to the XIAP-BIR3 domain determined by fluorescence polarization assay (see Figure 1). 

To predict the experimental ligand binding affinities, we chose to use the MM-PBSA method. Firstly, short molecular dynamics (MDs) simulations of 50 ns (NAMD 2.13) using all-atom CHARMM36m force field for protein and CGENFF for ligands were carried out for each tested XIAP-BIR3/ligand complex. During these simulations, the possible impact of different additional parameterizations, as the repartitioning of the hydrogen mass (HMR) [23] and the effect of parametrization of cation *π*-interactions for tryptophan, tyrosine and phenylalanine residues (WYF) [24], was tested. In HMR parametrization, the heavy atom mass is redistributed on the bound hydrogen atoms, making it possible to slow down the high-frequency movements of the macromolecule and increase the simulation time step. The cation-π interaction is a noncovalent binding force occurring between the aromatic ring, providing both a negative electrostatic potential surface and cations through electrostatic interaction. It was highlighted that cation-π interactions can play an important role in protein–ligand recognition. That is why we wanted to evaluate the influence of these two additional parametrizations in our study. In this way, each complex was simulated four times: (i) without any additional parametrization, (ii) with HMR, (iii) with WYF and (iv) with both WYF and HMR additional parametrizations. Finally, MM-PBSA methodology was applied on each simulation to predict ligand binding free energy $∆G_{MM-PBSA}$.

First, protein and ligand stability in the XIAP-BIR3 binding site in all simulations was checked visually and then through the Root Mean Square Deviation (RMSD) calculations. Visually, we observed that, during some simulations, the XIAP-BIR3 *N* and *C* termini fluctuated a lot (see Appendix A). Thus, to assess the stability of the binding site in each simulation, we aligned the protein conformations excluding both termini (without residues 248–258 for *N*-terminus and without residues 336–352 for *C*-terminus) and counted the backbone atom RMSD of the protein core (residues 259–335—see Appendix A). The RMSD did not exceed 1Å average in all simulations, thus confirming the XIAP-BIR3 binding site’s stability. To check the ligand stability during each simulation, a protein alignment without both termini was applied at first, and then the ligand RMSDs were calculated. On average, the ligand RMSDs were smaller than 2Å in all simulations (see Appendix A), thus confirming the ligand stability in the simulated complexes (with only the exception of ligand in complex 5C7C simulated with WYF parametrization—see Appendix A). During this simulation, the 1-methyl- 3-dimethyl-6-chloroindoline moiety of the ligand moves from its initial position around 15 ns, and it is positioned rather outside the XIAP-BIR3 binding site at the end of the simulation (RMSD = 3.97 ± 1.81Å).

Then, MM-PBSA analyses were carried out on the generated MD trajectories. Firstly, we estimated $∆G_{MM-PBSA}$ values without taking into account the entropic term, −T∆S, as usually applied in the literature (Table 1). Among the four parametrizations, the simulation without any additional parametrization usually applied in theoretical studies has given rather poor correlations (*r* = 0.56884 see Figure 2). On the other hand, the simulations with HMR and HMR + WYF additional parametrizations made it possible to predict ligand affinities in the correct order and with very good correlation compared to the experimental data. The best correlation between experimental and predicted ∆G was observed for HMR parametrization (*r* ≈ 0.98 see Figure 2A). Even if the correlation with HMR additional parametrization was very satisfactory, predicted $∆G_{MM-PBSA}$ values showed a significant difference (of about 30 kcal/mol) compared to the measured experimental data. We therefore decided to evaluate the contribution of the entropic part −*T*∆*S*, and we tested two methods described in the literature: the normal mode (NM) [25] method and the interaction entropy method (IE) [26,27]. The calculated $∆G_{MM-PBSA}$ values, including the entropic contribution for each tested simulation protocol, are summarized in Table 1. We observed that the adding of the entropic member calculated using the NM decreased the prediction accuracy in all four protocols. The non-tested protocol with NM entropic term predicted the correct order of ligand affinities. For all of the tested protocols, we obtained a satisfactory correlation (*r* ≤ 0.81, Figure 2B). In contrast, the $∆G_{MM-PBSA}$ values, which were calculated using the IE method, gave overall better results than those obtained using NM. A new MD simulation with the HMR additional parametrization produced a very good correlation (*r* = 0.97436; Figure 2C).

In conclusion, the HMR additional parametrization produces the best correlation between predicted and experimental free energies when the entropic term is not taken into account (Figure 2). Moreover, this parametrization also maintains a very good performance when the entropic term calculated using the IE method is included (*r* = 0.98264 without entropic term compared to *r* = 0.97436 with entropy from IE). Moreover, the predicted $∆G_{MM-PBSA}$ values with the entropic term calculated using the IE method are closer to the experimental $∆G_{exp}$ (Table 1); the differences between the predicted and experimental data were decreased to 10 kcal/mol.

We also probed the influence of the simulation temps on the correctness of the MM-PBSA prediction by computing $∆G_{MM-PBSA}$ values at 10 ns and 25 ns of the simulations (see Appendix A). We were able to identify certain trends. In the prediction without the entropic term, the lengthening of the simulations decreased the prediction accuracies for the simulations with any additional parametrization and with WYF additional parametrization, but it increased the exactness in the simulations with HMR additional parametrization. The same trend was also detected for the predictions with the entropic temp calculated using the IE method, except for the simulations without additional parametrization, for which the lengthening of the simulations increased the prediction accuracies this time. Nevertheless, in the case of predictions with the entropic term computed using NM, we did not succeed in deriving rules.

Next, we tested the prediction performance protocols on a ligand not belonging to the synthetic ligand category. We chose to use the AVPI tetrapeptide of Smac. To be able to compare the predicted free energies for the synthetic ligands to that of AVPI, the CGENFF force field was also applied to parameterize the AVPI tetrapeptide. As the XIAP-BIR3 3D structure co-crystallized with only AVPI tetrapeptide is not available in the PDB databank, we generated this complex by using the docking (see Appendix A) and submitted it to four MD simulation protocols, like for the previous complexes with synthetic ligands. The prediction accuracy of the binding free energy for the AVPI complex was also successful in the simulations with HMR additional parametrization, without the entropic term, as well as with the IE entropic term (Table 1). Very good results were also observed for the HMR + WYF additional parametrization protocol without the entropic term. This confirms that the use of the HMR additional parametrization significantly increases the exactness of the binding free-energy prediction in the MM-PBSA strategy. The inclusion of the entropic term calculated using the IE method maintains the correct accuracy for the HMR additional parametrization protocol, generating predicted values that are closer to the experimental ones.

To annotate the key interactions established between the studied ligands and the XIAP-BIR3 domain, contact maps were calculated for all simulated complexes, as well as their dynamical evolution along MD trajectories (see Appendix A). Afterward, we chose to focus our analysis on the 5M6L complex. Indeed, the 5M6L ligand is the most affine one, and we analyzed the contribution of each residue of XIAP-BIR3 to the ligand binding. To do so, an average ligand-binding free energy per residue was calculated from the MD simulation with HMR additional parametrization (Figure 3). The polar and non-polar contribution to binding free energy par residue was then analyzed separately (Figure 3C,D).

The analysis results showed that seven residues contribute to the ligand binding in the XIAP-BIR3 active site with a |ΔG| > 3.0 kcal/mol: Thr308, Glu314, Trp323, Leu307, Asp309, Trp310 and Gly306. Among these residues, the major polar contribution, calculated as the sum of electrostatic interaction energy and the polar solvation term, was assigned to the Glu314 (−8.0 ± 8.9 kcal/mol) and Thr308 (−5.6 ± 2.1 kcal/mol). These two residues, indeed, engage hydrogen bonds with the ligand: Glu314 side chain interacts through a salt bridge with one hydrogen atom of the protonated nitrogen of the piperazine ring of the ligand, and Thr308 interacts through its backbone carbonyl group with the second hydrogen atom. Thr308 established a second hydrogen bond by its backbone nitrogen atom with the ligand-carbonyl group. The occupation of these three hydrogen bonds at the course of the trajectory is globally in the same order, 87.4% for Glu314, 89.0% for the H-bond with Thr308 backbone nitrogen and 84.1% for the H-bond with Thr308 backbone oxygen atom (see Appendix A). In the course of the dynamics, the fluctuations of the ligand position in the XIAP-BIR3 binding groove occasionally caused the ligand to approach Asp309 (−1.7 ± 6.4 kcal/mol). Indeed, in 28.9% of the trajectory, a hydrogen bond is also formed between the protonated nitrogen atom of the piperazine ring of the ligand and Asp309 main chain carbonyl group. 

The analysis of the non-polar contribution (sum of van der Waals interaction energy with the non-polar solvation term) revealed several residues as stabilizing the ligand binding through non-polar contacts, and the most important contributions were detected for Trp323 (−7.9 ± 0.8 kcal/mol), Leu307 (−6.6 ± 0.8 kcal/mol), Thr308 (−5.3 ± 1.0 kcal/mol), Gly306 (−3.3 ± 0.5 kcal/mol), Trp310 (−3.0 ± 0.5 kcal/mol) and Asp309 (−2.9 ± 1.1 kcal/mol). Interestingly, Thr308 and its neighboring residue Asp309 contribute to the ligand binding through polar and non-polar interactions.

The published structural and mutation data on the XIAP-BIR3 complex determined the following residues as being critical in binding the Smac *N*-terminus: Glu314, Gln319 and Trp323 [12]. The residues annotated in our study showed that the synthetic ligands interact through the same residues. However, the Gln319 contribution to the synthetic ligand binding is less important (−2.6 ± 1.3 kcal/mol) according to our study.

### 2.2. Construction and Validation of a 3D Pharmacophore of Synthetic XIAP-BIR3 Inhibitors

In the final step, we built a representative 3D pharmacophore for the synthetic ligands binding to the XIAP-BIR3 binding groove. Indeed, the pharmacophore approach has become one of the important ligand-based tools in in silico methodologies for drug research and development process. The interest of this approach is to highlight essential motifs for the ligand–target interaction, which can then be applied to guide the design of new ligands. The LigandScout software [28] used in this study allowed us to generate the 3D pharmacophore either using only the ligand structures (ligand-based approach) or using the target structure in addition to the ligand structure (structure-based pharmacophore) [29]. In our study, in order to take into account the essential structural motifs for the interaction with the target, we generated XIAP-BIR3 pharmacophores by using the structure-based approach.

Our 3D pharmacophore model was built from 5M6L (X-ray structure co-crystallized with the most affine synthetic ligand to date). To generate the 3D pharmacophore, we exploited not only the crystallographic complex structure but also the previously generated MD trajectory simulated with HMR additional parametrization. We built one pharmacophore from the X-ray complex structure and another from the 2500 structures representing the entire MD trajectory (see Appendix A). The pharmacophore built from the X-ray structure had one hydrophobic and one aromatic–hydrophobic feature more and one hydrogen bond acceptor at least compared to the pharmacophore generated from MD trajectory (see Appendix A). The two pharmacophores were then aligned and merged, and the resulting pharmacophore comprised 8 features and 22 exclusion volumes (see Figure 4). Among the eight pharmacophoric features, LigandScout annotated four hydrophobic (H) and one aromatic–hydrophobic features (HAr), one hydrogen bond acceptor (HBA) and two hydrogen bond donors (HBD). The identified features of the pharmacophore reflect the ligand interaction with the XIAP-BIR3 binding site through Lys297, Thr308, Asp309, Glu314, Trp323 and Tyr324 (Figure 4). Thr308, Asp309, Glu314 and Trp323 residues already emerged as crucial from our free-energy decomposition per residue (Figure 3). In addition, LigandScout software also highlighted Lys297 and Tyr324 as important anchoring residues by positioning the hydrophobic features in their proximity. For Lys297 and Tyr324, we detected via our MM-PBSA analyses their dominant contribution non-polar one, which is in good agreement with the proposed hydrophobic characters of these residues. However, these two residues’ contributions to the global ligand-binding free energy were rather moderate: Lys297 (−2.4 ± 4.1 kcal/mol) and Tyr324 (−1.7 ± 0.8 kcal/mol).

To assess the generated 3D pharmacophore accuracy to discriminate active and inactive compounds, we used it to screen our test chemical library (see Materials and Methods Table 2) composed of 173 active compounds and 5417 inactive compounds. A good pharmacophore model would be able to identify a significant portion of active compounds and as few inactive ones as possible. Thus, an efficient screening should have high sensitivity, specificity, enrichment factor and area under the ROC curve values. The screening on our testing library showed that the pharmacophore n°1 performed rather weakly in terms of its sensitivity value (see Table 3). The sensitivity was only 2.9% when we overlaid all pharmacophoric features on the library ligands and 30.6% when we allowed the software to randomly discard one feature during alignment on the ligands. In view of these results, we decided to optimize it by omitting one or more features in order to test the impact on the overall screening performance. In view of the MM-PBSA results, we first tested the influence of features linked to Lys297 (first optimization round). We generated three pharmacophores, starting by deleting one (H4) of the two hydrophobic features linked to Lys297 (pharmacophore n°2, see Figure 4). Next, we removed the second remaining hydrophobic feature (H3), and we conserved the hydrophobic–aromatic feature (HAr) (pharmacophore n°3 on Figure 4). Finally, we removed the hydrophobic–aromatic feature (HAr) instead of the hydrophobic feature H3 (pharmacophore n°4; see Figure 4).

We observed that the removal of any of the three Lys297-related features did not notably improve the performance of the pharmacophore when the alignment of all pharmacophore features on the chemolibrary ligands was imposed: the sensitivity increased only by 1.1–6.9% (pharmacophore n°2–n°4; Table 3). In contrast, we noted that the sensitivity performances improved significantly for each tested pharmacophore when the screening allowed for the random exclusion of one feature during the alignment on ligands (pharmacophore n°1–n°4, Table 3). For example, the sensitivity increased from 9.8% to 77.5% for the pharmacophore n°4. Among the four tested pharmacophores in the first round of optimization, the n°4 one produced the best overall performance. We therefore selected it for the second optimization round. To perform this second optimization round, we analyzed, in detail, the discarded features in the screening (allowing for the random exclusion of one feature during the chemical-library alignment on pharmacophore n°4). We thus identified that the most frequently omitted feature among the six initial ones was the H1 hydrophobic feature. In view of these results, we generated a pharmacophore n°5 without the hydrophobic feature H1 (therefore composed of only five features). The library screening against this pharmacophore without the exclusion of any feature during alignment on the library revealed a very good performance (see Table 3). Altogether, the suppression of the H1 hydrophobic feature was unexpected because it was one of the features located near Trp323. Indeed, Trp323 is a residue highlighted as a one interacting strongly with the ligand in our per-residue free-energy analysis (non-polar contribution). To clarify the role of every part of the ligand in the interaction with the XIAP-BIR3 binding site, we carried out a decomposition of the interaction energy per ligand group (see Figure 5). The results confirmed the outcome of pharmacophore optimization. Indeed, we observed that Trp323 interacts strongly with 2,3-dihydro-pyrrolo[3,2-6]pyridine scaffold (G1 on Figure 5) and with carbonyl-linker (G3, specially with its hydrophobic CH_2_ group) and rather weakly with the morpholine part (G5 in Figure 5). Thus, the suppression of a feature placed on this morpholine will not affect the synthetic ligand interaction modelling with the XIAP-BIR3 binding site and, in particular, with Trp323. This analysis also highlighted that Tyr324 only plays a minor role in ligand binding (see Figure 5). The H2 presence of the hydrophobic feature of our pharmacophore is rather related to the interaction with Trp323 than with Tyr324.

In conclusion, this cross-analysis pointed out as important groups for ligand binding to XIAP-BIR3 the protonated piperazine moiety (G4) and carbonyl-linker (G3), with the two groups forming hydrogen bonds between ligand and XIAP-BIR3. We also highlighted the important contribution of the 2,3-dihydro-pyrrolo[3,2-6]pyridine scaffold (G1) and 4-fluoro-benzyl moiety (G2), which interact mainly in a hydrophobic manner. We observed that the G1 group interacts mostly with Trp323, while the G2 group interacts simultaneously with several residues on a moderate level: Lys297, Gly306, Leu307 and Thr308 (see Figure 5B). These results lead us to believe that this pocket of the BIR3 binding site hosting the group G2 is not completely filled by published synthetized ligands.

Interestingly, comparing XIAP-BIR3 residues to cIAP1-BIR3 and cIAP2-BIR3 residues revealed crucial residues that bind to synthetic ligands (see Figure 5C). Only three residues are not conserved between XIAP and cIAP1-2: Lys297/Asp, Thr308/Arg and Asp309/Cys. It can therefore be concluded that targeting these three residue side chains could increase the ligand selectivity for XIAP versus cIAP1-2. Among them, the Lys297 side-chain interaction through pharmacomodulations of the G2 group seems to be the most promising way to improve the selectivity.

A pharmacophore for XIAP-BIR3 inhibitors has already been published in the literature [30]. Opo and co-workers also generated their pharmacophore by using LigandScout software, but from the 5OQW X-ray structure, one of the structures used in our MM-PBSA study. Unlike us, their pharmacophore was generated by also considering the co-crystallized water molecules in the XIAP-BIR3 binding site, and it is composed of twelve features: four hydrophobic features, one positive ionizable group, three hydrogen bond acceptors, five hydrogen bond donors and fifteen exclusion volumes. The performance of our optimized pharmacophore n°5 is of the same level compared to the AUC value, and it is even better from the point of view of the enrichment factor, 20.4, with respect to 10.0. Indeed, our simpler 3D pharmacophore, with only five features, represents the chemical groups necessary for synthetic ligands to bind in the XIAP-BIR3 binding groove. 

## 3. Conclusions

The aim of this study was to establish a computational tool that would allow us to design and/or optimize the XIAP-BIR3 selective synthetic ligands. Firstly, we demonstrated that the MM-PBSA approach for free-energy calculation from molecular dynamics predicts ligand-binding affinities in the right order (CHARMM36 force field and HMR additional parametrization). The inclusion of the entropic term calculated using the interaction energy methodology maintains good predictability of the binding affinities, and the free-energy predicted values tend to be closer to the experimental ones. Our study showed that the key residues for synthetic ligand binding are Thr308 and Glu314. They both form hydrogen bonds with the ligand protonated nitrogen of the piperazine ring and with the ligand carbonyl-linker part. Therefore, the presence of double H-bond donors, as well as the H-bond acceptor on ligands, is required to bind to XIAP-BIR3. Indeed, the 3D pharmacophore for XIAP-BIR3 inhibitors, built and optimized by us, reflects these observations. Among the retained five features that are necessary for the binding of synthetic ligands to XIAP-BIR3, we revealed two hydrogen-bond donors and one hydrogen-bond acceptor. Nevertheless, these H-bonds are formed either through Thr308 main-chain atoms or through Glu314 side-chain carboxyl groups. Glu134 is a conserved residue (Glu/Asp) in the IAPs family and does not allow us to reach XIAP-BIR3 ligand selectivity. 

We also highlighted that the hydrophobic interactions are established principally by Trp323, Leu307, Thr308, Gly306, Trp310, Asp309 and Lys297. Indeed, our 3D pharmacophore contains two hydrophobic features, 2,3-dihydro-pyrrolo[3,2-6]pyridine scaffold with Trp323) and 4-fluoro-benzyl moiety with various residues: Lys297, Gly306, Leu307 and Thr308. Among these residues, only three are not conserved between XIAP and cIAPs (Lys297, Thr308 and Asp309), which can be exploited to design XIAP-BIR3 versus cIAP1-2 specific small inhibitory compounds. According to our study, the most promising strategy seems to be the pharmacomodulations of the 4-fluorobenzyl moiety interacting with the Lys297 side chain.

## 4. Materials and Methods

### 4.1. Structure Preparation

During this study, X-ray structures of the following complexes with XIAP-BIR3 retrieved from the PDB database were used (Table 4): 5C7C [20], 5M6M [21], 5OQW [22] and 5M6L [21]. Then, to calculate the experimental value of ligand binding free energy $∆G_{exp}$, the following equation was applied:(1)$$∆G_{exp}=RT\times \ln{IC}_{50}$$
where the gas constant value was *R* = 1.985 8775 × 10^−3^ kcal·K^−1^·mol^−1^ and *T* = 303, 15 K.

As the XIAP-BIR3 complex with Smac AVPI tetra peptide was not solved to date, to prepare this complex, we chose to apply the docking strategy. The AVPI tetra peptide, which was retrieved from the PDB structure of XIAP-BIR3 co-crystallized with full SMAC (PDB ID: 1G73 [12]), was docked into the X-ray crystal structure 5C7C of XIAP-BIR3, without waters and bounded synthetic ligand. The docking was carried out using Glide software (Schrödinger^®^) with the default parameters [31]. The protein structure was prepared before using Schrödinger Protein Preparation Wizard (PrepWizard) [31]. The Glide score was used to evaluate the generated AVPI poses, and the pose with the best score was selected as a starting structure for MD simulations.

### 4.2. Molecular Dynamics Simulations

All dynamics simulations were performed using NAMD 2.13 [32]. The all-atom CHARMM36m forcefield [33,34] was used for the XIAP-BIR3 protein, and CGENFF [35] was used for ligands. Two additional force fields were applied in this order: (i) hydrogen mass repartition (HMR) [23] and (ii) additional parametrizations of *π*-cation interaction for tryptophan, tyrosine and phenylalanine (WYF) residues [24]. The starting systems were generated by the CHARMM-GUI server [36]. The three Cys residues, Cys300, Cys303 and Cys327, coordinating the zinc cation were parametrized as anionic cysteine (CYM), and the four histidine residues were modeled along the PropKa suggestion [37,38]: 320 and 346 as HSD, and 302 and 343 as HSE. Each system was solvated using the TIP3P explicit water model [39] within a rectangular box; the box size ensured that the simulated complex was at a minimum distance of 10 Å from the edge. To neutralize the total charge system, 0.15 M of KCl was added. The vacuum dielectric constant was used during all calculations. Cubic periodic boundary conditions were applied to the systems by using the IMAGE algorithm. The applied cutoff distance was 16 Å, and van der Waals interactions were truncated using a force-switching function between 10 and 12 Å. The Particle Mesh Ewald (PME) was used to calculate long-range electrostatic interactions [40]. The SHAKE algorithm was applied to restrain all bonds involving hydrogen atoms.

Each complex was subjected to 4 different dynamics simulations to assess the impact of additional parameters of the force field (HMR and WYF). Firstly, each underwent energy minimization in 10,000 steps, with harmonic restraints applied on heavy atoms (1 kcal/mol/Å^2^ force constant for backbone atoms and 0.5 kcal/mol/Å^2^ force constant for sidechain atoms), followed by 10,000 steps of minimization without any restraints. Next, the minimized systems were heated to 303.15 K, and the dynamics were temperature-equilibrated during 250 ps via heating reassignment under NVT conditions, with harmonic restraints applied to heavy atoms. Finally, the systems ran freely for 50 ns under NPT conditions, with a time step of 2 fs for the simulations without HMR additional parametrization and 4 fs with HMR parametrization. Langevin dynamics with a damping coefficient of 1 ps^−1^ was used to maintain the system temperature, and the Nosé–Hover–Langevin piston method was used to control the pressure at 1 atm. Production trajectories were saved every 20 ps, and subsequent analyses were performed using the CHARMM program version c40b2 [41].

### 4.3. MM-PBSA Prediction

To estimate the binding free energy of a ligand to a receptor, the MM-PBSA method was applied. MM-PBSA calculates the binding free energy as the sum of the classical enthalpic contributions (bound, van der Waals and electrostatic energies), the solvation-free energies and the entropic contribution [42]. In our protocol, we simulated only the complex and the ensemble of data for the free receptor and ligand for each snapshot created by removing the appropriate atoms. Then, the binding free energy was calculated using the following equation [43,44]:(2)$$∆G=\langle E_{inte}\rangle +\langle G_{polar}\rangle +\langle G_{non-polar}\rangle -T\langle S\rangle$$
where the first term, *E_inte_*, is the standard Molecular Mechanics ligand–receptor interaction energy calculated from electrostatic and van der Waals interaction energies; and $G_{polar}$ and $G_{non-polar}$ are the polar and non-polar contributions to the solvation free energies, respectively. $G_{polar}$ was determined by solving the Poisson–Boltzmann equation, whereas the $G_{non-polar}$ term was estimated from a linear relation to the solvent accessible surface area (SASA), using the equation 0.00542 × SASA + 0.92. The value of $G_{polar}$ and $G_{non-polar}$ in each trajectory snapshot was calculated as follows:(3)$$G_{polar}=G_{Complex}^{PB}-G_{Recptor}^{PB}-G_{Ligand}^{PB}$$
(4)$$G_{non-polar}=G_{Complex}^{SASA}-G_{Recptor}^{SASA}-G_{Ligand}^{SASA}$$

To calculate all contributions to the MM-PBSA CHARMM, homemade scripts were applied.

For the last term, entropic contribution, three approaches were tested: firstly, (i) the entropic term was omitted, and then the entropic term was approximated using either (ii) normal mode analysis of harmonic frequencies calculated at the Molecular Mechanics level [25] or using the recently published approach (iii) interaction entropy that investigates the entropy change upon binding [26,27]. The interaction entropy (IE) contribution to binding free energy is defined as follows:(5)$$-T∆S=k_{B}T\times \ln\langle \exp\left\{\frac{∆E_{inte}}{k_{B}T}\right\}\rangle$$
where $∆E_{inte}=E_{inte}-\langle E_{inte}\rangle$ is the fluctuation of the receptor–ligand interaction energy. The ensemble average of $\langle E_{inte}\rangle$ was extracted from MD simulations.

### 4.4. 3D Pharmacophore Building

The 3D pharmacophore was generated using the 3D structure of human XIAP-BIR3 co-crystallized with a synthetic inhibitor (PDB ID code 5M6L [21]) and using the structural data resulting from its MD simulations with additional HMR parametrization. LigandScout software [28] was applied for the detection and interpretation of crucial interaction patterns between XIAP-BIR3 and the ligand. LigandScout extracts and interprets ligands and their macromolecular environment from PDB files and automatically creates and visualizes advanced 3D pharmacophores. Besides the features representing the interaction of the ligand with the target (as hydrogen bond donor, hydrogen bond acceptor, hydrophobic group, etc.), we also generated our pharmacophore exclusion volumes.

In order to compare the pharmacophore performance during the optimization rounds, we referred to the evaluation metrics, such as sensitivity, specificity, enrichment factor and area under the ROCs (Receiver Operating Characteristics) curve. The purpose of these statistical tests is to clearly differentiate the pharmacophore performances from each other. Sensitivity determines the model’s ability to retain active molecules. In contrast, specificity assesses the model’s ability to reject inactive molecules. The enrichment factor (EF) is the ratio between the number of active ligands aligned with the pharmacophore and the total number of ligands. The enrichment factor therefore measures the enrichment of active compounds in the hit list compared to a pure random selection. A method that is superior to random selection results in an EF value greater than 1. The area under the ROC curve (AUC), on the other hand, evaluates the probability that, for two screened ligands, one active and the other inactive, the score value is higher for the active than for the inactive. The AUC value between 0.5 and 1 is mandatory for a model to be validated (AUC = 1 perfect prediction; AUC = 0.5 random prediction).

These tests are defined by the following equations [45]:(6)$$Sensitivity=\frac{N\;selected\;actives}{N\;total\;actives}=\frac{TP}{A}$$
(7)$$Specificity=\frac{H\;discarded\;inactives}{H\;total\;inactives}=\frac{TN}{I}$$
(8)EF=TPHtotAD
where TP is the number of active ligands aligned with the pharmacophore, TN is the number of discarded inactive ligands and FP is the number of inactive ligands aligned with the pharmacophore. Moreover, Htot = (TP + FP) is the total number of aligned ligands, A is the number of active ligands in the initial dataset, I is the number of inactive ligands in the initial dataset and D is the total number of ligands in the initial dataset.

In general, an efficient screening should have high values of sensitivity, specificity, enrichment factor and area under the ROC curve. During the screening, we applied an alignment of all features on the ligands and the extraction mode based on the most stable ligand energy conformation. For each pharmacophore, we carried out two screenings: (i) in the first one, we imposed that all features of the 3D pharmacophore model had to be aligned to ligands; and (ii) in the second screening, the software could randomly discard one feature during the alignment on ligands.

### 4.5. Testing Chemical Library

To test our 3D pharmacophore performance, we created a validation dataset. To build it, we first detected all compounds annotated with XIAP-BIR3 biological activities in CHEMBL database [46,47]. From this dataset, the redundant ligands and the peptide ligands or peptide-mimetic ones were deleted. This resulted in a dataset of 487 synthetic XIAP-BIR3 ligands. Then, it was necessary to determine a *cutoff* (threshold) to classify the ligands in terms of biological activity (active/inactive). As the number of ligands available for XIAP-BIR3 was limited, we applied the approach that is usually used in the literature for targets that are not perfectly annotated [48]. In this strategy, a pIC_50_ (${pIC}_{50}=-{log}_{10}{IC}_{50}$) value of 6 is usually used as a *cutoff.* All molecules with a pIC_50_ value greater than 6 are considered active, and those with a value under 6 are considered inactive. Moreover, molecules with a pIC_50_ value between 4.5 and 6 are excluded from the dataset (see Table 2). This led us to a dataset with a ratio of active to inactive ligands of 1/1.3 (i.e., 173 active ligands with respect to 218 inactive ligands). This ratio was not sufficient to obtain statistically significant values during the screening and correctly assess its performance.

Thus, an increase in the number of inactive compounds in our dataset was necessary to improve the screening performance, and to do so, we included XIAP-BIR3 decoys from the DUD-E database [48] (compounds were assumed to be inactive and built from fragments of active molecules). For our target, there were 5199 decoys available that we added to our starting dataset, resulting in a library of 5686 compounds. This library was then used to evaluate the performance of our 3D pharmacophores (see Table 2).

The 3D structure of library compounds was generated and protonated at pH = 7.4, using LigandPrep tool of Schrödinger^®^ software [31].

## Figures and Tables

**Figure 1 molecules-28-05155-f001:**
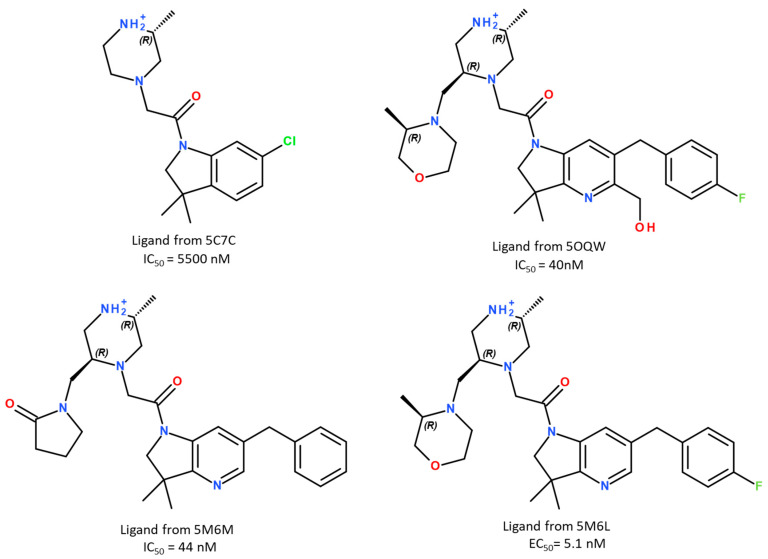
The chemical structures of the ligands used in this study.

**Figure 2 molecules-28-05155-f002:**
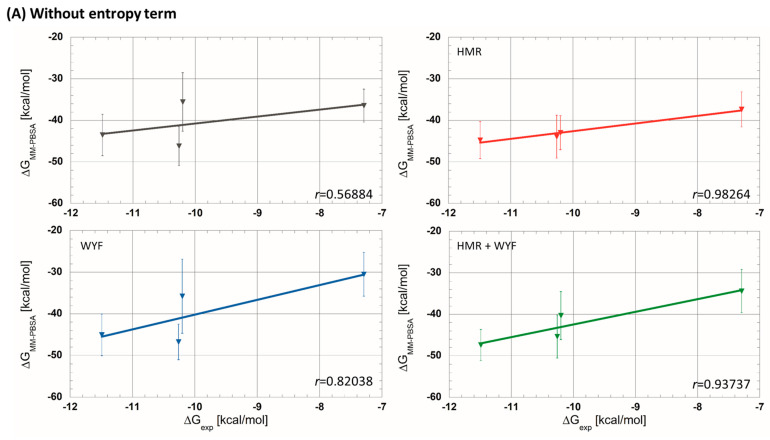
Experimental free-energy correlations with predicted ones using MM-PBSA approaches (**A**) without entropic term, (**B**) with entropic term calculated using the normal mode strategy and (**C**) with entropic term calculated using interaction entropy method (gray curves—without additional parametrization; red—HMR; blue—WYF; green—HMR and WYF additional parametrizations).

**Figure 3 molecules-28-05155-f003:**
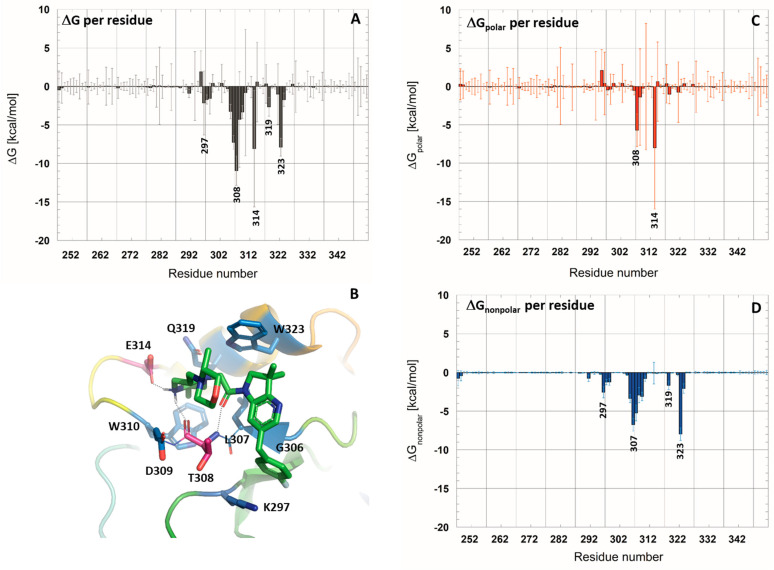
Calculated free-energy contribution per residue to ligand binding in 5M6L complex from molecular dynamics simulation with HMR additional parametrization. (**A**) Δ*G_MM-PBSA_* per residue without entropic term. (**C**,**D**) Polar and non-polar decomposition of the Δ*G_MM-PBSA_* per residue. (**B**) XIAP-BIR3 binding site view with strongly contributing residues colored in red (polar contribution) and blue (non-polar contribution).

**Figure 4 molecules-28-05155-f004:**
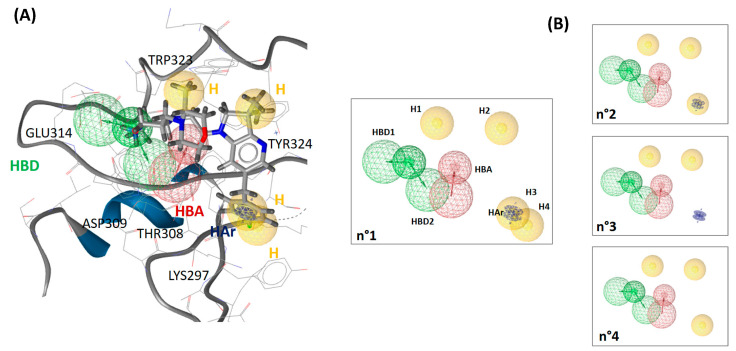
(**A**) 3D pharmacophore, resulting from the fusion of the crystal structure pharmacophore and the MD trajectory pharmacophore, aligned with the ligand of the 5M6L structure. The exclusion volumes were omitted for clarity. The essential residues for the ligand binding are annotated in the figure. (**B**) The three derived 3D pharmacophores tested during the validation.

**Figure 5 molecules-28-05155-f005:**
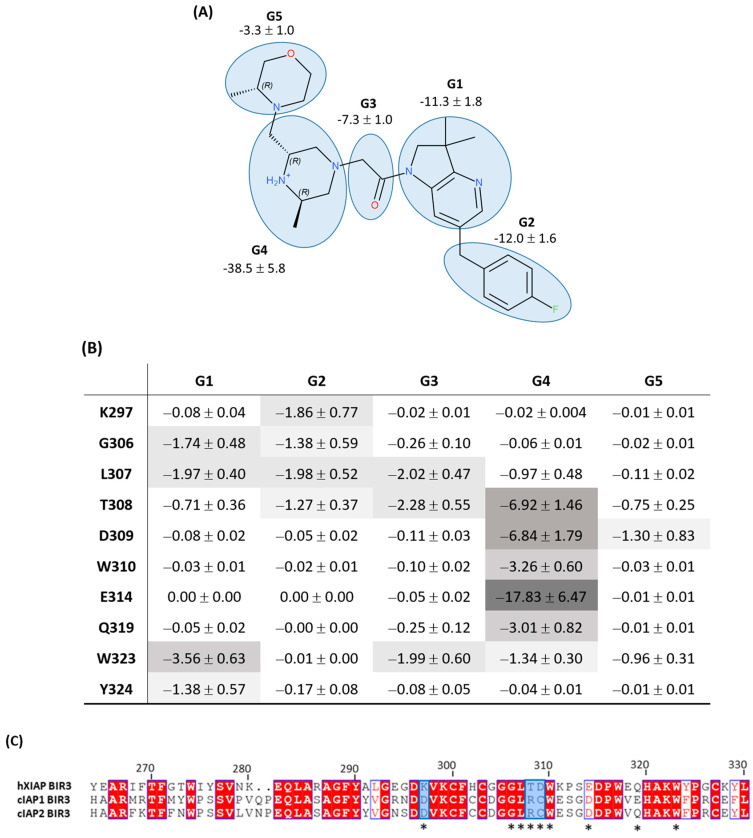
(**A**) Decomposition of ligand/XIAP-BIR3 interaction energy per ligand group in kcal/mol. (**B**) Interaction energy cross-analysis in kcal/mol per ligand group and per residue for the 10 residues revealed from MM-PBSA free-energy analysis or from pharmacophore construction. (**C**) XIAP/cAIP1-2 BIR3 domain sequence alignment, * annotate the residues interacting with the ligand.

**Table 1 molecules-28-05155-t001:** Experimental binding free energy, Δ*G_exp_*, and calculated ones using the MM-PBSA method, Δ*G_MM-PBSA_*, for different parametrizations of the force field.

(**a**) Without entropic term.									
	**Δ*G_exp_* (kcal/mol)**	**Δ*G_MM-PBSA_* (kcal/mol)**							
		**Without Additional Parametrization**		**HMR**		**WYF**		**HMR + WYF**	
5C7C	−7.29	−36.5 ± 4.0		−37.4 ± 4.2		−30.5 ± 5.3		−34.4 ± 5.2	
5M6M	−10.20	−35.6 ± 7.0		−43.0 ± 4.1		−35.8 ± 8.9		−40.3 ± 5.8	
5OQW	−10.26	−46.2 ± 4.7		−43.9 ± 5.1		−46.8 ± 4.3		−45.4 ± 5.2	
5M6L	−11.49	−43.5 ± 5.0		−44.7 ± 4.4		−45.1 ± 5.0		−47.4 ± 3.7	
AVPI	−8.91	−46.2 ± 4.9		−39.5 ± 6.6		−47.5 ± 9.2		−38.2 ± 5.9	
(**b**) With entropic term calculated using normal mode analysis of harmonic frequencies (NM) or using the interaction entropy method (IE).									
	**Δ*G_exp_* (kcal/mol)**	**Δ*G_MM-PBSA_* (kcal/mol)**							
		**Without Additional Parameterization**		**HMR**		**WYF**		**HMR + WYF**	
		**NM**	**IE**	**NM**	**IE**	**NM**	**IE**	**NM**	**IE**
5C7C	−7.29	−25.6 ± 2.8	−10.9 ± 4.4	−26.5 ± 3.0	−17.6 ± 4.6	−18.3 ± 3.2	−17.5 ± 5.4	−23.7 ± 3.6	−8.1 ± 6.3
5M6M	−10.20	−18.7 ± 3.7	− 8.9 ± 8.8	−27.5 ± 2.6	−24.9 ± 4.4	−17.7 ± 4.4	−4.6 ± 12.1	−23.5 ± 3.4	−12.1 ± 6.3
5OQW	−10.26	−29.9 ± 3.0	−28.0 ± 5.0	−27.0 ± 3.1	−24.2 ± 5.7	−30.8 ± 2.8	−34.0 ± 4.4	−30.1 ± 3.5	−25.8 ± 5.4
5M6L	−11.49	−27.9 ± 3.2	−24.2 ± 5.6	−29.2 ± 2.9	−25.6 ± 4.8	−28.8 ± 3.2	−30.0 ± 5.1	−32.3 ± 2.5	−26.1 ± 4.4
AVPI	−8.91	−28.7 ± 3.0	−33.2 ± 5.0	−23.0 ± 3.8	−22.8 ± 7.0	−32.2 ± 6.2	−31.7 ± 9.9	−21.0 ± 3.2	−19.5 ± 6.2

**Table 2 molecules-28-05155-t002:** Characteristics of the library used for 3D pharmacophore validation.

Active/inactive cutoff in pIC_50_	6
Total number of ligands	5590
Number of active molecules	173
Number of inactive molecules	5417
Molecules with 6 > pIC_50_ > 4.5 not retained	96
Active/inactive ratio	1/31
Number of ligands retained in the testing library	5494

**Table 3 molecules-28-05155-t003:** Statistical screening results with the testing chemical library for each different pharmacophore (TP = selected compounds from 173 active ones; TN = selected compounds from 5417 inactive ones).

Pharmacophore	Number of Omitted Features during Ligand Alignments	Sensitivity	Specificity	Enrichment Factor	AUC at 1.5%	AUC at 100%
n°1	0	TP = 5 2.9%	TN = 5417 100%	32.3	1	0.51
	1	TP = 53 30.6%	TN = 5413 99.9%	30.0	1	0.65
n°2	0	TP = 7 4.0%	TN = 5417 100%	32.3	1	0.52
	1	TP = 97 56.1%	TN = 5411 99.9%	30.4	1	0.78
n°3	0	TP = 7 4.0%	TN = 5417 100%	32.3	1	0.52
	1	TP = 96 55.5%	TN = 5155 95.2%	8.7	1	0.76
n°4	0	TP = 17 9.8%	TN = 5416 100%	30.5	1	0.55
	1	TP = 134 77.5%	TN = 5056 93.3%	8.7	1	0.87
n°5	0	TP = 136 78.1%	TN = 5338 98.5%	20.4	1	0.89

**Table 4 molecules-28-05155-t004:** X-ray structures used in this study summary.

PDB ID	Resolution (Å)	R-Free	IC_50_ (nM)	Δ*G_exp_* (kcal/mol)
5C7C [20]	2.32	0.286	5500	−7.29
5M6M [21]	2.37	0.233	44	−10.20
5OQW [22]	2.31	0.246	40	−10.26
5M6L [21]	2.61	0.263	5.1 *^a^*	−11.49
AVPI	Docking into 5OQW	-	320	−8.91

*^a^* Published value was an EC_50_ value.

## Data Availability

Not applicable.

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
