# Peer review of "Computational Tool to Design Small Synthetic Inhibitors Selective for XIAP-BIR3 Domain"

_molecules, 2023, doi:10.3390/molecules28135155_

Round 1

Reviewer 1 Report

Farag et al., present an interesting characterization of ligands for the BIR3 domain of XIAP. I believe the work is presented in a compelling way, yet there are some comments/concerns which I will develop henceforth:

There are some editing or grammatic errors in the text; for example in line 152: "free energies when the entropic term is not take into account"

Figures 2 and 3 in the main text are difficult to follow due to the scale and overall format. Similar comments also apply to figures in the supplementary information.

The steps followed for additional parametrization are not clear. I understand that these were made with CHARMM-GUI was that the case?

Considering the observed trends for ligands' RMSD could the authors provide interaction histograms or concurrence plots? see: https://contact-map.readthedocs.io/en/latest/examples/nb/concurrences.html

I mostly agree with the process and rationale behind the development of pharmacophore model. Still, I'm curious on the potential effect of the imbalance in the dataset. I'm not really sure, but was there any control for overfitting in this scenario? I say this mostly due to the overall performance which may suggest said phenomenon. I understand that this is more common in machine learning contexts; however, perhaps the inactive compounds share significant similarities and thus a bias exist to correctly identify them. I mention this due to use of the DUD-E dataset, as it is now known decoy compounds have biases.

This is more a suggestion but considering the overall performance and findings I wonder if the authors could include a comparison with MM/GBSA as similar works have proven very valuable, more so in the context of achieving selectivity.  For reference consider: 10.1021/acs.jcim.7b00347

No comment

Author Response

There are some editing or grammatic errors in the text; for example in line 152: "free energies when the entropic term is not take into account"

Figures 2 and 3 in the main text are difficult to follow due to the scale and overall format. Similar comments also apply to figures in the supplementary information.

Thank you for your careful review. The grammatical errors were corrected in the manuscript and the figures 2 and 3 were improved in the main text.

The steps followed for additional parametrization are not clear. I understand that these were made with CHARMM-GUI was that the case?

Considering the observed trends for ligands' RMSD could the authors provide interaction histograms or concurrence plots? see: https://contact-map.readthedocs.io/en/latest/examples/nb/concurrences.html

A paragraph explaining the additional parametrisations using in this study was added in the main text as well as graphs representing key interacting residues of XIAP-BIR3 with ligands and their dynamical evolution along MD trajectories for all simulated complexes (Figure S5).  

I mostly agree with the process and rationale behind the development of pharmacophore model. Still, I'm curious on the potential effect of the imbalance in the dataset. I'm not really sure, but was there any control for overfitting in this scenario? I say this mostly due to the overall performance which may suggest said phenomenon. I understand that this is more common in machine learning contexts; however, perhaps the inactive compounds share significant similarities and thus a bias exist to correctly identify them. I mention this due to use of the DUD-E dataset, as it is now known decoy compounds have biases.

Indeed, it is a risk to include compounds that have not been biologically assessed in the dataset, but our initial dataset of biologically evaluated compounds from ChEMBl has a ratio of active to inactive ligands too low, it was of 1/1.3 (i.e. 173 active ligands with respect to 218 inactive ligands). This ratio was not sufficient to obtain statistically significant values during the pharmacophore screening and so asses correctly its performance. So, an increase of the number of inactive compounds in our data set was necessary to improve the performance evaluation of our pharmacophore, and to do it we included the XIAP-BIR3 decoys from the DUD-E database.

This is more a suggestion but considering the overall performance and findings I wonder if the authors could include a comparison with MM/GBSA as similar works have proven very valuable, more so in the context of achieving selectivity.  For reference consider: 10.1021/acs.jcim.7b00347

I'm sorry but the deadline for the correction is too short to carry out the MM-GBSA analyses. For this comparison to make sense, we need to use the same software and the same parameters, which means to develop scripts in CHARMM to run MM/GBSA analysis.

Reviewer 2 Report

This Text require minor language editing. 

examples:

Introduction 

Escape from apoptosis is one of the major causes of cancer development and progression. It also contributes to chemoresistance [1].   line 29

a member of the inhibitor of 32 apoptosis family of proteins (IAPs), 

lines 32,33

This Text require minor language editing. 

examples:

Introduction 

Escape from apoptosis is one of the major causes of cancer development and progression. It also contributes to chemoresistance [1].   line 29

a member of the inhibitor of 32 apoptosis family of proteins (IAPs), 

lines 32,33

Author Response

Thank you for your careful review. The grammatical errors were checked in the manuscript.

Reviewer 3 Report

Jana Sopkova-de Oliveira Santos et. al reported an interesting computational design for optimizing the pharmacophore for designing synthetic ligands for the XIAP-BIR3 domain Although interesting, the work, these shortcomings must be adequately addressed for the acceptance of the manuscript

1.     The authors mentioned,  the Δ??????? calculated using the IE method gave overall the better results than these obtained using NM, and new  MD simulations with the HMR additional parametrization produced a very good correlation (r= 0.97436, Figure 2C). Can the authors brief out a little more about it for more clarification?

2.     The authors quoted. As the 3D structure of XIAP-BIR3 co-crystallized with only AVPI tetrapeptide is not available in the PDB databank, we generated this complex using the docking. The Authors should include the relevant figures and data for the same.

1. The level of English of the manuscript is not sufficient for an academic publication. Proofreading by an expert or native speaker might be helpful.

2. There are grammatical mistakes and a few typo errors which needs attention.

Author Response

Thank you very much for your remarks.

  1. The authors mentioned, the Δ???−???? calculated using the IE method gave overall the better results than these obtained using NM, and new MD simulations with the HMR additional parametrization produced a very good correlation (r= 0.97436, Figure 2C). Can the authors brief out a little more about it for more clarification?

A paragraph explaining the additional parametrisations using in this study (HMR as WYF) has been added in the main text.

  1. The authors quoted. As the 3D structure of XIAP-BIR3 co-crystallized with only AVPI tetrapeptide is not available in the PDB databank, we generated this complex using the docking. The Authors should include the relevant figures and data for the same.

A detailed figure of the docked AVPI peptide into XIAP-BIR3 was added in SI Figure S4.

The grammatical errors were corrected in the manuscript.

Reviewer 4 Report

The study focused on designing and optimizing selective non-peptidic or non-peptidomimetic antagonists for XIAP-BIR3, a critical regulator of apoptosis. The goal of this research was to provide a computational tool for creating and improving specific synthetic ligands for XIAP-BIR3. The researchers found that integrating the entropic term into the MM-PBSA technique using the CHARMM36 force field enhanced the prediction of ligand binding affinities. Thr308 and Glu314 were identified as key residues for ligand binding because they generated hydrogen bonds with the ligand's piperazine ring and carbonyl-linker portion. The researchers also created and optimized a 3D pharmacophore for XIAP-BIR3 inhibitors, emphasizing the significance of two hydrogen bond donors and one hydrogen bond acceptor. The study is well-organized, well-written, and contains several important details and findings. The paper deserves to be published after adderessing the minor revisions below.

The abstract lacks contextual information: Overall, the abstract summarizes the research aims, methodologies, and conclusions. However, there are several areas where clarity and precision may be enhanced. While the abstract gets right to the point of the work, it would be helpful to add some context on the significance of the XIAP protein and its role in apoptosis regulation. This would assist readers in comprehending the significance and relevance of the research.

Insufficient context for the research objective: While the introduction briefly mentions the study's focus on designing selective non-peptidic and non-peptidomimetic antagonists of XIAP-BIR3, it could provide more context regarding the limitations of current approaches and the potential impact of developing such antagonists. This would better emphasize the significance of the research.

Author Response

The abstract lacks contextual information: Overall, the abstract summarizes the research aims, methodologies, and conclusions. However, there are several areas where clarity and precision may be enhanced. While the abstract gets right to the point of the work, it would be helpful to add some context on the significance of the XIAP protein and its role in apoptosis regulation. This would assist readers in comprehending the significance and relevance of the research.

Insufficient context for the research objective: While the introduction briefly mentions the study's focus on designing selective non-peptidic and non-peptidomimetic antagonists of XIAP-BIR3, it could provide more context regarding the limitations of current approaches and the potential impact of developing such antagonists. This would better emphasize the significance of the research.

Both the abstract and the introduction chapter were reformulated and completed in the new version of the main text.

Reviewer 5 Report

The current study employed the MM-PBSA approach to compute free energy and make predictions about ligand binding affinities. The authors propose potential strategies for developing selective small inhibitory compounds targeting XIAP-BIR3 over cIAP1-2.

Overall, this study is intriguing and will make a valuable contribution to the research community working on in-silico methods. I have a few suggestions for minor revisions that I believe could enhance the existing methods and approaches proposed in this work.

  1. This study is particularly interesting and will complement existing research on binding free energy calculations using MM-GBSA and PBSA methods. I am curious if the authors have also attempted to calculate the MM-PBSA values from shorter simulations. If not, it might be worth exploring this aspect, as there are limited studies focusing on it. Examining the effect of simulation length on binding free energy would be a valuable addition to the investigation.

  2. Please review the sentences on lines 42-44 and 205 for clarity and accuracy.

Author Response

  1. energy calculations using MM-GBSA and PBSA methods. I am curious if the authors have also attempted to calculate the MM-PBSA values from shorter simulations. If not, it might be worth exploring this aspect, as there are limited studies focusing on it. Examining the effect of simulation length on binding free energy would be a valuable addition to the investigation.

Thank you very much for your review. The table showing deltaG values for shorter simulations was added (SI Table S1 and S2) and a short discussion of these results was included in the main text.

  1. Please review the sentences on lines 42-44 and 205 for clarity and accura

The sentences on lines 42-44 and 205 were reformulated

Round 2

Reviewer 1 Report

The work shows substantial improvement.

Personally I still find figures 2 and 3 difficult to follow due to the size and scaling.

Beyond that, most of my concerns have been addressed. I final suggestion would be the inclusion of a couple of lines mentioning the need of further validation using other methods, particularly molecular dynamics.

There are some minor errors remaining. I suggest a careful and through revision

Author Response

Thank you very much for this second review. We have enlarged Figures 2 and 3.

Regarding your request to add the sentence “ I final suggestion would be the inclusion of a couple of lines mentioning the need of further validation using other methods, particularly molecular dynamics.”

We would just like to note that the two approaches applied in our study used molecular dynamics simulations as a starting point. Our MM-PBSA binding free energies were calculated for each complex and for each tested parametrization of force field from molecular dynamics trajectories of 50 ns (we used 1250 frames). This is why for each delta G prediction by MM-PBSA we have able also to estimate uncertainty (standard deviation) of the prediction. Each value of ΔGMM-PBSA presented on Figures 2 and 3 is accompanied by errors bars.

To generate the 3D pharmacophore we also used in addition to the crystallographic complex a molecular dynamics simulation of 50 ns.

English has been reviewed by native English person present actually in the laboratory.